# Exosomes Derived from Adipose Mesenchymal Stem Cells Promote Diabetic Chronic Wound Healing through SIRT3/SOD2

**DOI:** 10.3390/cells11162568

**Published:** 2022-08-18

**Authors:** Yue Zhang, Xiaozhi Bai, Kuo Shen, Liang Luo, Ming Zhao, Chaolei Xu, Yanhui Jia, Dan Xiao, Yan Li, Xiaowen Gao, Chenyang Tian, Yunchuan Wang, Dahai Hu

**Affiliations:** Department of Burns and Cutaneous Surgery, Xijing Hospital, Fourth Military Medical University, 127 Changle West Road, Xi’an 710032, China

**Keywords:** chronic wounds, angiogenesis, ADSC, exosomes, SIRT3, oxidative stress, inflammation

## Abstract

Chronic wounds resulting from diabetes are a major health concern in both industrialized and developing countries, representing one of the leading causes of disability and death. This study aimed to investigate the effect of adipose mesenchymal stem cell-derived exosomes (ADSC-exos) on diabetic wounds and the mechanism underlying this effect. The results showed that ADSC-exos could improve oxidative stress and secretion of inflammatory cytokines in diabetic wounds, thereby increasing periwound vascularization and accelerating wound healing. At the cellular level, ADSC-exos reduced reactive oxygen species (ROS) generation in human umbilical vein endothelial cells (HUVECs) and improved mitochondrial function in a high-glucose environment. Moreover, the Western blot analysis showed that the high-glucose environment decreased Sirtuin 3 (SIRT3) expression, while exosome treatment increased SIRT3 expression. The activity of superoxide dismutase 2 (SOD2) was enhanced, and the level of inflammatory cytokines was decreased. Further, SIRT3 interference experiments indicated that the effects of ADSC-exos on oxidative stress and angiogenesis were partly dependent on SIRT3. After SIRT3 was inhibited, ROS production increased, while mitochondrial membrane potential and SOD2 activity decreased. These findings confirmed that ADSC-exos could improve the level of high-glucose-induced oxidative stress, promote angiogenesis, and reduce mitochondrial functional impairment and the inflammatory response by regulating SIRT3/SOD2, thus promoting diabetic wound healing.

## 1. Introduction

Approximately 15% of diabetic patients worldwide face slow wound healing and/or diabetic foot ulcers, and amputation of the affected foot rates are as high as 14–24% [1]. The main causes of these issues comprise difficulty in vascular reconstruction, peripheral neuropathy, and persistent inflammation [2,3]. Vascular function plays an important role in chronic diabetic wound healing, providing nutrients and oxygen for wound healing throughout the process [4]. Angiogenesis is mainly mediated by vascular endothelial cells. However, a high-glucose environment induces the production of high reactive oxygen species (ROS) levels, leading to mitochondrial defects, apoptosis, and inflammation in vascular endothelial cells [5]. The ultimate manifestations are impaired vascular function, insufficient angiogenesis, restricted extracellular matrix remodeling, and delayed diabetic wound healing [6]. Despite efforts to understand the causes of impaired chronic wound healing in diabetes, the underlying molecular mechanisms are not fully understood. Effective treatment of chronic diabetic wounds is still an urgent clinical problem.

Mesenchymal stem cell-derived exosomes (MSC-exos) can promote wound healing in various ways [7]. They can promote granulation tissue formation and angiogenesis by preventing endothelial progenitor cell (EPC) senescence and inhibiting ROS and inflammatory cytokine expression [8]. Additionally, they can promote M2 macrophage polarization by affecting the nuclear factor kappa B (NF-ĸB) signaling pathway, reducing the level of wound inflammation and oxidative stress-related proteins, thus promoting wound healing [9]. As such, MSC-exos have emerged as important targets for chronic diabetic wounds.

SIRT3, a mitochondrial deacetylase, could improve oxidative stress and protect vascular function by regulating mitochondrial function and reducing ROS production in endothelial cells [10]. A previous report showed that a reduction in SIRT3 expression in diabetic mice inhibits the expression of vascular endothelial growth factor (VEGF) and impairs antioxidant capacity, resulting in delayed wound healing [11]. This effect is related to impaired mitochondrial function and increased necroptosis. SIRT3 overexpression attenuates mitochondrial oxidative stress and reduces vascular inflammation, which, in turn, protects vascular function [12].

While the role of adipose mesenchymal stem cell-derived exosomes (ADSC-exos) and SIRT3 in chronic diabetic wounds and treatment outcomes is somewhat understood, the relationship between ADSC-exos and SIRT3, whether ADSC-exos can influence oxidative stress and inflammation by regulating SIRT3, and which functional molecules play a role in promoting diabetic wound healing have not been reported. In this study, we examined the role of ADSC-exos and SIRT3 in oxidative stress and angiogenesis in wound healing of diabetic mice and HUVECs.

## 2. Materials and Methods

### 2.1. Isolation, Culture, and Identification of ADSCs

Human normal subcutaneous adipose tissues were acquired from patients undergoing selective liposuction at Xijing Hospital (Xi’an, China). The patients provided informed consent prior to the study. Then, the adipose tissues were fragmented and digested with 1 mg/mL collagenase type I (Gibco; Thermo Fisher Scientific, Waltham, MA, USA) for 60 min. After being filtered through a 100 μm mesh, the isolated cells were obtained by centrifugation at 200 g for 5 min. The resultant cell pellet was resuspended in human ADSC Expansion Media (Gibco; Thermo Fisher Scientific, Inc.) and cultured at 37 °C. The medium was changed every 3 days. Then, the cells were examined by flow cytometry (BD FACSAria ™ III system; BD Pharmingen) to determine the expression of CD31, CD34, CD90, and CD105. Osteogenic and adipogenic induction of ADSCs was performed as previously described [13]. Mature osteocytes secreting calcium phosphate were detected 24 days after the osteogenic induction. Lipid droplets were identified after the adipogenic induction for approximately 21 days under an FSX100 microscope (Olympus, Tokyo, Japan).

### 2.2. Exosome Isolation and Identification

Exosomes were extracted from ADSC supernatants as previously described [14]. The isolated exosomes were characterized by transmission electron microscopy (TEM). Western blotting was performed to measure the expression of the exosomal markers CD9 and CD63. The average concentration of exosomal proteins was 2 μg/μL, as measured by a BCA protein assay kit. The isolated exosomes were stored at −80 °C for subsequent use. Isolated exosomes were labeled with PKH26 (Sigma-Aldrich, St Louis, MO, USA) and added to HUVECs. After 12 h of culturing, DAPI was used to stain the nuclei. Then, immunofluorescence signals were observed with an FSX100 microscope.

### 2.3. Animal Models

Healthy db/db mice at approximately 7 weeks of age were purchased from Cavens (Changzhou, China). Blood glucose (BG) was measured with a One-Touch blood glucometer (Johnson, New Brunswick, NJ, USA) via the tail vein. All mouse BG levels were higher than 16.7 mmol/L. After the mice were anesthetized with 2% isoflurane, a single, round full-thickness skin wound with a diameter of 10 mm was created by surgical scissors. Three days later, phosphate-buffered saline (PBS) or ADSC-exos (200 µg) were administered by subcutaneous injection into the wound for three consecutive days. The size of the wounded area was measured and recorded by digital photographs on days 0, 1, 3, 5, 7, 10, and 14. The mice were sacrificed at the indicated times, and wound tissues were harvested for subsequent analysis. Wound healing rate (%) on Y axis = (wound area on day 0-wound area on day X)/wound area on day 0 × 100%. There were at least six mice in each experimental group.

### 2.4. Haematoxylin and Eosin (HE) and CD34 Staining

Mouse skin tissue from the edge of the wound was collected, fixed with 4% paraformaldehyde, dehydrated, embedded, and mounted. Then, the sections were stained with HE, and the FSX100 microscope (Olympus, Tokyo, Japan) was used to observe the sections.

Staining for CD34 (Santa Cruz Biotechnology, Santa Cruz, CA, USA) was used to assess vasculature in the wounds. The most vascularized area of the tissue section was examined as previously described [15]. More than 10 nonoverlapping areas were examined to determine the microvessel density (MVD) of the skin tissue.

### 2.5. Measurement of Oxidative Stress in Skin Tissue

The level of malondialdehyde (MDA) in the skin samples was assessed with the thiobarbituric acid method according to the manufacturer’s instructions (Beyotime, Shanghai, China). The 2,2’-azino-bis (3-ethylbenzthi-azoline-6-sulfonic acid, ABTS) method was used to determine the total antioxidant capacity (T-AOC, Beyotime, Shanghai, China). SOD activity was measured with the WST-8 [5-(2,4-disulfophenyl)-3-(2-methoxy-4-nitrophenyl)-2-(4-nitrophenyl)-2H-tetrazolium] method (Beyotime, Shanghai, China) according to the manufacturer’s protocol.

### 2.6. Cell Culture, Treatment, and Transfection

HUVEC lines were obtained from the China Centre for Type Culture Collection (CCTCC). The cells were cultured in DMEM (Gibco, Grand Island, NY, USA) supplemented with 10% fetal bovine serum (Corning, NY, USA), 100 U/mL penicillin, and 100 µg/mL streptomycin and placed in an incubator, at 37 °C, with 5% CO_2_. The cells were grown to 80% confluence for the following experiments. HUVECs were exposed to high glucose (33 mM) for 24 h to mimic the conditions of the diabetes mellitus (DM) model, followed by subsequent analysis.

HUVECs were transfected with SIRT3 small interfering RNA (siRNA) purchased from Santa Cruz Biotechnology (Santa Cruz, CA, USA) to silence SIRT3. Lipofectamine 2000 reagent (Invitrogen, Life Technologies, Grand Island, NY, USA) was used according to the manufacturer’s instructions for 12 h in a serum-free medium, while negative control siRNA (NC) was used as a control. The efficiency was measured by reverse transcription-polymerase chain reaction (RT-PCR) at 24 h after the transfection.

### 2.7. HUVECs Proliferation, Migration, and Tube Formation Assays

Cell proliferation, migration, and tube formation assays were performed to evaluate the effects of ADSC-exos on HUVECs under high-glucose conditions. The method was performed as previously described [16]. HUVECs were cultured in 96-well plates at a density of 2 × 103 cells/well. CCK-8 (Cell Counting Kit-8) solution (10 μL/well) was added at 0, 24, and 48 h and then incubated for 2 h, at 37 °C. A microplate reader was used to measure the optical density (OD) at a wavelength of 450 nm.

A 10 µL pipette tip was used to induce a scratch in HUVECs in different groups in a six-well plate at a density of 1 × 106 cells/well. Wound closure was photographed at 0 h, 24 h, and 48 h by the FSX100 microscope (Olympus, Tokyo, Japan). HUVECs were seeded on 1 mL of DMEM at a density of 2 × 104 cells/well in a 24-well plate coated with 200 μL of Matrigel (BD Biosciences, Bedford, MA, USA) per well. After being incubated at 37 °C for 10 h, tube formation was examined using FSX100 microscope, and total tube length and the number of branch points per well were quantified by ImageJ software. This experiment was performed three times for each condition.

### 2.8. Measurement of Intracellular ROS

According to the instructions of the ROS assay kits (Beyotime, Shanghai, China), HUVECs were incubated with 2 μmol/L dihydroethidium (DHE), at 37 °C, for 30 min, and the relative levels of intracellular ROS were measured by flow cytometry (BD Pharmingen) according to the manufacturer’s instructions.

### 2.9. MMP Assay

HUVECs were incubated with mitochondrial membrane potential ΔΨm (JC-1) staining solution according to the instructions (Beyotime, Shanghai, China) for 30 min and then washed twice with JC-1 buffer. Images were taken by the FSX100 microscope (Olympus, Tokyo, Japan). The ratio of red-to-green fluorescence represents the Mitochondrial Membrane Potential (MMP). JC-1 show red fluorescence in normal cells and green fluorescence when cells injury, thus the MMP declines. 

### 2.10. Quantitative RT-PCR

Total RNA was extracted by TRIzol reagent (Invitrogen, Waltham, MA, USA) according to the instructions. Then, RNA was reverse transcribed into cDNA using the Prime Script RT Reagent kit (Takara, Japan). Then, the cDNA was mixed with the SYBR Green quantitative PCR (qPCR) mixture (Takara, Japan) for amplification. A Bio-Rad PCR instrument (Bio-Rad, Hercules, CA, USA) was used to perform PCR. The fold changes were calculated by the comparative delta–delta cycle threshold method. PCR primers are listed in Table 1.

### 2.11. Western Blot Analysis

An amount of 20 μg protein was loaded into each well. Then, cellular proteins were separated by sodium dodecyl sulfate-polyacrylamide gel electrophoresis (SDS-PAGE) and then transferred to polyvinylidene fluoride (PVDF) membranes (Millipore, Billerica, MA, USA). After being blocked in 5% nonfat dry milk for 2 h, the membranes were incubated with primary antibodies against AC-SOD2 (Abcam, Cambridge, UK), SOD2 (Abcam, Cambridge, UK), and SIRT3 (Abcam, Cambridge, UK), at 4 °C, overnight. All primary antibodies were diluted at 1:1000. Glyceraldehyde 3-phosphate dehydrogenase (GAPDH) (1:3000; Abcam, Cambridge, UK) was used as a control. After the membranes were washed three times, they were incubated with HRP-conjugated anti-rabbit immunoglobulin (Ig)G secondary antibodies (1:3000, Boster, Wuhan, China) for 1 h, at 37 °C. Using an ECL detection system (Millipore, Burlington, MA, USA), the intensity of protein expression on the membrane was measured on a FluorChem FC system (Alpha Innotech, San Jose, CA, USA). The results were analyzed by ImageJ software and normalized against GAPDH.

### 2.12. Statistical Analysis

All data were analyzed using SPSS 17.0 software (IBM Inc., Chicago, IL, USA) and are presented as the mean  ±  standard error of the mean. Student’s t test was used for comparisons between two groups, and analysis of variance (ANOVA) was used for multigroup comparisons. A *p* value of < 0.05 was considered statistically significant. The specify acronyms were in Appendix A.

## 3. Results

### 3.1. Extraction and Identification of ADSCs and Exosomes

ADSCs were extracted from adipose tissue and subjected to adipogenic and osteogenic induction (Figure 1A). Figure 1B shows high expression of the ADSC surface markers CD90 and CD105 and almost no expression of CD31 or CD34, which confirmed the successful isolation of ADSCs. ADSC-exos were extracted by differential centrifugation, and a typical bilayer cup-shaped membrane structure was observed by electron microscopy (Figure 1C). The Western blot analysis showed positive expression of exosomal surface markers CD63 and CD9. These results indicated that ADSCs and their exosomes were successfully isolated.

### 3.2. ADSC-Exos Promote Angiogenesis and Accelerate Wound Closure

A diabetic wound mouse model was used to verify the effect of ADSC-exos on healing. Fasting blood glucose (FBG) was measured via the tail vein of db/db mice and exceeded 16.7 mmol/L. A full-thickness skin injury with a diameter of 1 cm was created on the backs of the mice, and 100 μL of PBS or 200 µg of exosomes were subcutaneously injected around the wound for 3 consecutive days. The wound area in the exosome treatment group was always smaller than that in the control group, and the wound surface in the exosome group was basically closed on the 14th day (Figure 2A,B). HE staining was performed to assess the degree of re-epithelialization. On day 14, the amount of intact new epidermis and dermis increased in the exosome group compared to the PBS group (Figure 2C). The new blood vessels provided nutrients and oxygen to the wound area and were essential for wound repair. On day 14, more CD34+ cells were observed in the exosome group than in the control group. About 6 nonoverlapping areas were examined to determine the number of blood vessels of the skin tissue and found that blood vessel density was significantly higher than that in the control group, indicating good promotion of angiogenesis (Figure 2D, Appendix A). 

MDA, total antioxidant capacity (T-AOC), and SOD kits were then used to examine oxidative stress-related indicators in the wounds in each group. The results showed that on the 7th day, the level of MDA in the exosome group decreased, while the levels of SOD and T-AOC increased (Figure 2E). This result indicated that exosomes significantly improved oxidative stress in diabetic wounds. In addition, mRNA expression of VEGF in the exosome group was significantly increased. It was also observed that VCAM and inflammatory cytokines interleukin (IL)1, IL6,tumor necrosis factor (TNF)-α and MCP-1 was decreased in the exosome group (Figure 2F). These data proved that exosome treatment could improve oxidative stress and inflammation in wounds, promote angiogenesis, and accelerate wound healing.

### 3.3. ADSC-Exos Promote the Proliferation, Migration, and Angiogenesis of HUVECs

We first examined whether exosomes could be internalized by cells to determine the effect of exosomes on vascular endothelial cells. Exosomes labeled with green fluorescent dye PKH26 were transferred to the perinuclear region of vascular endothelial cells after 12 h of incubation (Figure 3A). The proliferation of HUVECs was examined in response to 12.5 μg/mL, 25 μg/mL, 50 μg/mL, and 100 μg/mL exosomes. The CCK-8 results showed that the proliferation rate of treated HUVECs significantly increased with increasing time and exosome concentrations. The proliferation rate of HUVECs treated with 100 μg/mL ADSC-exos for 48 h was the highest (Figure 3B). ADSC-exos increased HUVECs migration (Figure 3C) and angiogenesis (Figure 3D) under high-glucose conditions and increased the expression of the angiogenesis-related proteins Angiopoietin-1 (ANG1), Fetal Liver Kinase-1 (FILK1), and VEGF but decreased mRNA expression levels of endogenous angiogenesis inhibitors Vasohibin 1 (VASH1) and Thrombospondin-1 (TSP1). These results confirmed the potential of ADSC-exos to enhance angiogenesis in vascular endothelial cells in a high-glucose environment.

### 3.4. ADSC-Exos Reduce ROS Production, Protect Mitochondrial Function, and Promote SIRT3 Expression

The production of ROS in different cell groups was examined to further examine the protective mechanism of ADSC-exos on vascular endothelial cells. ROS levels were significantly increased in cells exposed to high glucose, and ADSC-exos pretreatment suppressed ROS production in these cells (Figure 4A). Subsequently, HUVECs were treated with 50 µg of ADSC-exos for 12 h and then exposed to high glucose (33 nM) for 12 h. Then, the protective effect of exosomes on mitochondria in HUVECs stimulated by high glucose was assessed by JC-1 staining. Compared with the control group, the high-glucose group had a significantly reduced MMP, while pretreatment with ADSC-exos prevented the loss of MMP and reduced mitochondrial damage (Figure 4B). Exosomes reduced the expression of HUVECs adhesion molecules VCAM and ICAM in the high-glucose environment and normalized inflammatory cytokine (IL1, IL6, and TNFα) and chemokine (MCP-1) levels (Figure 4C). The Western blot results proved the increased SOD2 protein expression and decreased Acetylated SOD2 (AC-SOD2) levels in the exosome group compared with the high-glucose group (Figure 4D). The expression of SIRT3 in HUVECs was significantly reduced by high glucose, while the addition of exosomes restored the protein levels of SIRT3 (Figure 4D). This result suggested that the SIRT3 pathway was damaged in HUVECs in the high-glucose environment, and this effect could be improved by exosomes.

### 3.5. ADSC-Exos Reduce ROS Production and Protect Mitochondrial Function through SIRT3

To investigate the role of SIRT3 in the antioxidative effects of ADSC-exos, we used SIRT3-specific siRNA and showed that when SIRT3 was silenced, the expression of SOD2 and deacetylation levels in the exosome group decreased (Figure 5A). Moreover, exosome-mediated reductions in intracellular ROS levels were attenuated (Figure 5B). JC-1 staining results showed that when SIRT3 was silenced, MMP was decreased in the exosome group, and the protective effect of exosomes on mitochondria was weakened (Figure 5C). According to the PCR results, SIRT3 interference attenuated the ADSC-exo-mediated decreases in inflammatory cytokines, ICAM-1, VCAM-1, IL1, IL6, TNFα, and monocyte chemoattractant protein (MCP1) (Figure 5D). These results indicated that ADSC-exos could protect against high glucose-induced injury in HUVECs by regulating SIRT3.

### 3.6. The Ability of ADSC-Exos Is Weakened When SIRT3 Is Silenced

Our group further investigated the role of SIRT3 in the angiogenic effects of ADSC-exos. When SIRT3 was silenced, the ability of exosomes to promote migration (Figure 6A) and angiogenesis (Figure 6B) was impaired. PCR results showed that when SIRT3 was silenced, exosome-mediated promotion of the mRNA expression of angiogenesis-related factors ANG1, FILK1, and VEGF in HUVECs was reduced, and the reduction in the mRNA expression of endogenous angiogenesis inhibitors VASH1 and TSP1 was abrogated (Figure 6C). These results demonstrated that ADSC-exos promoted angiogenesis in HUVECs through SIRT3 in the high-glucose environment.

## 4. Discussion

Chronic diabetic wounds are difficult to treat, and therapies are urgently needed to improve. Many studies have reported that mesenchymal stem cells are widely used in tissue repair and regeneration due to their strong self-renewal and multidirectional differentiation capabilities [17,18,19]. ADSCs can be easily obtained by minimally invasive techniques and have multilineage differentiation and self-renewal abilities. Additionally, these cells accelerate wound healing [20]. However, their survival rate is low after injection, and their therapeutic activity is poor. Furthermore, cell transplantation may trigger immune and inflammatory responses, affecting their clinical application [21]. ADSC-exos, which are important mediators of intercellular communication, can promote angiogenesis and regulate oxidative stress and inflammatory response [22,23]. In this study, our group successfully extracted ADSC-exos and identified them by TEM and Western blotting. Then, their role in the healing of diabetic wounds was investigated with the db/db diabetic mouse model of full-thickness skin injury on the back. ADSC-exos could promote angiogenesis, inhibit oxidative stress, reduce inflammation, and significantly accelerate wound healing in diabetic mice. Our findings undoubtedly provide new insights into the important role of ADSC-exos in chronic diabetic wounds.

Chronic diabetic wounds are difficult to heal, mainly due to dysregulation of the wound microenvironment resulting from the effects of high glucose, including oxidative stress, chronic inflammatory reactions, and microangiopathy [24]. Angiogenesis is involved in wound healing and is a critical factor necessary for the healing of chronic diabetic wounds. New blood vessels promote wound healing by transporting nutrients, oxygen, and growth factors to the injured site [25]. However, the high-glucose environment in diabetic patients leads to dysfunction in vascular endothelial cells, resulting in reduced angiogenesis, which is characterized by reduced blood vessel formation and decreased capillary density [26]. Our previous study has shown that under high-glucose conditions, in vitro HUVECs proliferation and migration were inhibited, and angiogenesis was impaired [27]. In the present study, we evaluated the effect of ADSC-exos on the proliferation, migration, and angiogenesis of HUVECs under high-glucose conditions. ADSCs-exos significantly enhanced the proliferation and migration of HUVECs in the high-glucose environment. Additionally, improved angiogenesis and more intact tubular structures were observed in HUVECs treated with ADSC-exos compared to the high-glucose group. The expression of angiogenesis-related factors confirmed that ADSC-exos could increase mRNA levels of angiogenesis factors ANG1 and FLK1 and reduce the levels of endogenous angiogenesis inhibitors VASH1 and TSP1. These data showed the potential of ADSC-exos to enhance vascular endothelial cell function and promote angiogenesis in the high-glucose environment.

The current study on the pathogenesis of diabetic vascular complications and vascular endothelial cell dysfunction has shown that hyperglycemia can promote the generation of ROS in endothelial cells, which, in turn, activates the protein kinase C pathway and/or the pathway forming advanced glycation end-products, thereby inducing endothelial cell injury [28]. However, the generation of high ROS levels can not only reduce endothelial cell activity through complex signaling pathways, but also induce the secretion of inflammatory factors, such as TNF-α and IL-6, and the expression of various adhesion molecules by activating the NF-κB signaling pathway, thereby promoting inflammation [5,29]. This inflammation will further exacerbate vascular endothelial cell dysfunction. Although a moderate number of free radicals promote wound healing, excessive ROS can inhibit the migration and proliferation of repairing cells, inhibit the synthesis of extracellular matrix, and ultimately delay wound healing [30]. Our study showed that ADSC-exos could improve oxidative stress and inflammation in diabetic wounds, and the high-glucose environment could increase HUVECs ROS production and promote the expression of adhesion molecules (ICAM and VCAM), MCP1, and inflammatory cytokines (TNF-α, IL-1, and IL-6) and aggravate the inflammatory microenvironment of the wound. Mitochondria are the main sites of ROS production, and mitochondrial structural damage or dysfunction can easily disrupt homeostasis and induce oxidative damage [31]. Noxious stimuli alter the permeability of mitochondrial membranes to reduce membrane potential, block the mitochondrial electron transport chain, inhibit cytochrome c production and adenosine triphosphate (ATP) levels, and accelerate ROS accumulation [32]. This study showed that the high-glucose environment could induce abnormal MMP and that ADSC-exos increased the MMP in HUVECs in the high-glucose environment, cleared the accumulated ROS, and reduced the expression of inflammatory cytokines and adhesion molecules.

SOD, which is also known as manganese-dependent SOD, belongs to the iron/manganese SOD family. SOD is a superoxide scavenging enzyme playing a crucial role in maintaining cellular redox balance [33]. Hyperacetylation of SOD2 resulted in its inactivation and increased oxidative stress [34]. Deacetylation of SOD2 at Lys68 is a necessary step for SOD2 activation, reducing the accumulation of ROS and preventing oxidative stress [35]. This study showed that ADSC-exos could increase the activity of SOD2 in diabetic wounds and decrease AC-SOD2 protein in HUVECs exposed to high glucose, demonstrating that ADSC-exos could reduce the accumulation of ROS in vascular endothelial cells in the high-glucose environment by influencing SOD2.

SIRT3 regulates mitochondrial function and reduces oxidative stress by deacetylating several mitochondrial proteins [35]. SIRT3 inhibition increases mitochondrial ROS production. In this study, ADSC-exos significantly increased SIRT3 protein levels in HUVECs exposed to high glucose. SIRT3 downregulation blocked the effect of ADSC-exos on the level of SOD2 deacetylation. In HUVECs cultured in the high-glucose environment, the positive regulation of ADSC-exos on oxidative stress, cellular activity, and mitochondrial dysfunction induced by high glucose was reversed when SIRT3 expression was silenced. SIRT3 siRNA (siSIRT3) reduced the level of SOD2 deacetylation in HUVECs despite exposure to ADSC-exos. With the accumulation of ROS, the expression levels of ICAM-1, VCAM-1, inflammatory cytokines (TNF-α, IL-1, and IL-6), and MCP1 were all enhanced. Moreover, the exosome-mediated improvements in HUVECs migration and angiogenesis in the high-glucose environment were partially abrogated. This finding indicated that SIRT3 was a key mediator by which ADSC-exos improve endothelial cell injury and dysfunction. Our findings are consistent with research showing that SIRT3 and SOD2 were reduced in dysfunctional EPCs in hypertensive patients, thereby impairing endothelial repair capacity. Knockout of SIRT3 inhibited the re-endothelization abilities of EPCs, while SOD2 induced mitoTEMPO, reduced mitochondrial damage, and rescued EPC function [10,36]. All in all, we clarified the relationship between ADSC-exos and SIRT3 and undoubtedly provided new insight into the important role of ADSC-exos in diabetic wound.

## 5. Conclusions

In conclusion, our results suggested that ADSC-exos could regulate the expression of SIRT3 and its downstream protein SOD2, which ameliorated vascular endothelial cell dysfunction caused by hyperglycemia by improving oxidative stress and inflammatory microenvironment, thereby promoting angiogenesis and further healing chronic diabetic wounds. The present study has some limitations. This study only preliminarily explored the potential mechanism by which ADSC-exos mediate SIRT3/SOD2 to improve chronic diabetic wounds. Future studies should focus on the molecular mechanism by which ADSC-exos regulate SIRT3, which will help identify the role of ADSC-exos in diabetic wounds.

## Figures and Tables

**Figure 1 cells-11-02568-f001:**
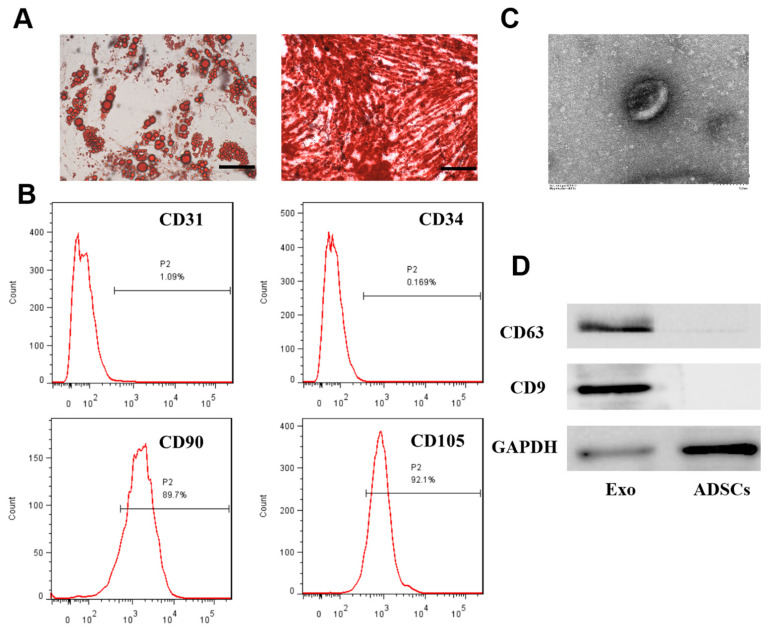
The identification of ADSCs and ADSCs-exos. (**A**) Adipogenic and osteogenic differentiation was examined by Oil Red O staining and Alizarin Red S staining (scale = 50 μm). (**B**) Analysis of ADSCs by flow cytometry. The results showed negative expression of CD31 and CD34 (1.09% and 0.169%, respectively) and positive expression of CD90 and CD105 (89.7% and 92.1%, respectively). (**C**) Analysis of ADSC-exos morphology by TEM (scale = 100 nm). (**D**) Western blot analysis of exosome markers (CD63 and CD9).

**Figure 2 cells-11-02568-f002:**
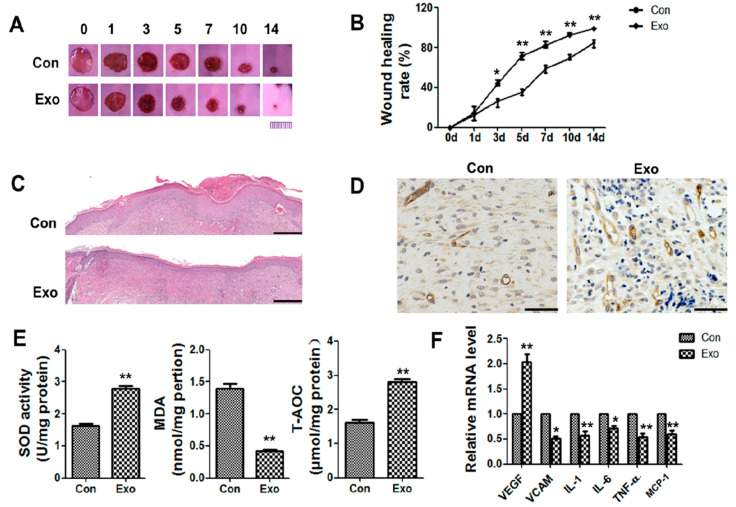
Effects of ADSC-exos on wound healing in db/db model mice. (**A**) Digital images of wound areas treated with PBS or ADSC-exos on days 0, 1, 3, 5, 7, 10, and 14 (scale = 1 cm). (**B**) Analysis of the wound closure rate. (**C**) HE staining of normal skin or wound tissue treated with ADSC-exos on day 14 (scale = 500 μm). (**D**) CD34 staining of wound margins on day 14 (scale = 50 μm). (**E**) The levels of SOD, MDA, and T-AOC on day 7 were analyzed by SOD, lipid oxidation (MDA), and T-AOC detection kits. (**F**) mRNA levels of VEGF, VCAM, IL1, IL6, TNF-α, and MCP-1 on day 3 were analyzed by qRT-PCR. The data are shown as the mean  ±  standard deviation (SD; * *p*  <  0.05, ** *p*  <  0.01).

**Figure 3 cells-11-02568-f003:**
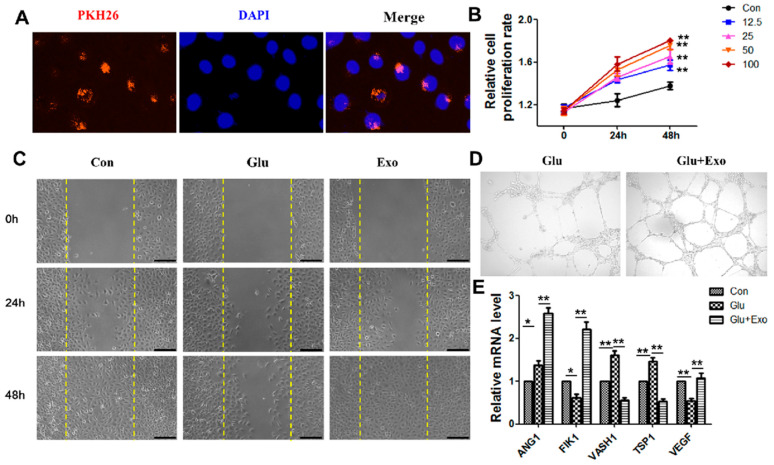
The effect of ADSC-exos on HUVECs. (**A**) Representative immunofluorescence images of PKH26-labelled exosomes in HUVECs. (**B**) The effect of ADSC-exo concentration on HUVECs activity was determined by CCK-8 assays. (**C**) The effect of ADSC-exos on HUVECs migration in a high-glucose environment was examined by scratch wound assays. The cells were divided into a normal control group, a high glucose control group, and an ADSC-exo-treated high glucose group. The images show the wound area in each group at 0 h, 24 h, and 48 h (scale = 250 μm). (**D**) Angiogenesis in different groups. The results showed that ADSC-exos enhanced angiogenesis in HUVECs. (**E**) mRNA levels of angiogenesis factors (ANG1, FILK1, and VEGF) and angiogenesis inhibitors (VASH1 and TSP1) were analyzed by qRT-PCR. The data are shown as the mean  ±  SD (* *p*  <  0.05, ** *p*  <  0.01).

**Figure 4 cells-11-02568-f004:**
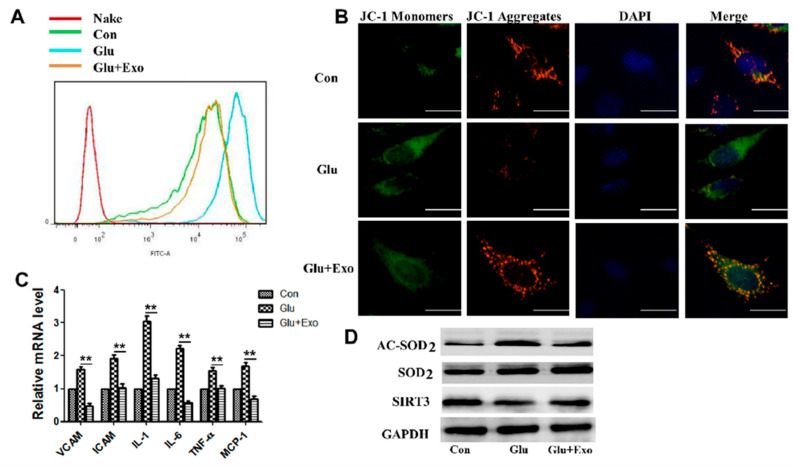
ADSC-exos reduce ROS production in HUVECs in a high-glucose environment, increase the MMP, reduce inflammatory cytokine levels, and promote SIRT3 expression. (**A**) Analysis of HUVECs ROS levels by flow cytometry. (**B**) JC-1 signal in HUVECs was examined by fluorescence confocal microscopy. The cells were labeled with DAPI to show the nucleus (blue) and stained with JC-1 to show the mitochondria. Double staining of cells with JC-1 is shown: green for J-monomers, red for J-aggregates (scale = 50 μm). The data represent three independent measurements. (**C**) mRNA levels of VCAM, ICAM, IL1, IL6, TNFα, and MCP-1 in HUVECs were analyzed by qRT-PCR. (**D**) Western blot results showing the protein levels of AC-SOD2, SOD2, and SIRT3 in HUVECs treated with ADSC-exos in the high-glucose environment. The data are shown as the mean  ±  SD (** *p*  <  0.01).

**Figure 5 cells-11-02568-f005:**
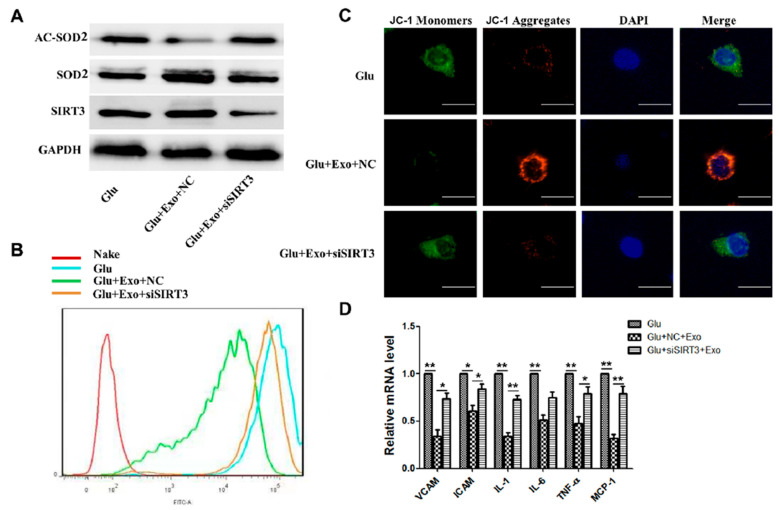
ADSC-exos exert their effects through SIRT3. (**A**) Western blot results showing HUVECs AC-SOD2, SOD2, and SIRT3 expression in the high-glucose group, high-glucose+exo+NC group, and high-glucose+exo+SIRT3 group. (**B**) HUVECs ROS levels were measured by flow cytometry. (**C**) HUVECs JC-1 signals were measured by fluorescence confocal microscopy (scale = 50 μm). (**D**) mRNA levels of VCAM, ICAM, and inflammatory cytokines (IL1, IL6, TNF-α, and MCP-1) were analyzed by qRT-PCR. The data are shown as the mean  ±  SD (* *p*  <  0.05, ** *p*  <  0.01).

**Figure 6 cells-11-02568-f006:**
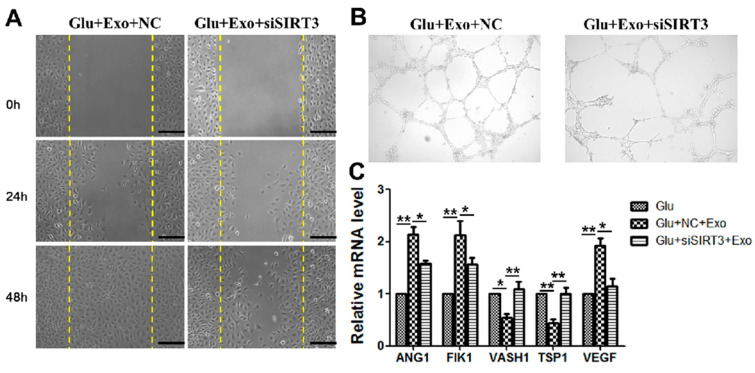
ADSC-exos promote the migration and angiogenesis of HUVECs via SIRT3. (**A**) The effect of SIRT3 on HUVECs migration was examined by scratch wound assays. The images show the wound area in each group at 0 h, 24 h, and 48 h (scale = 250 μm). (**B**) Angiogenesis in different groups of HUVECs. The results showed that siSIRT3 reduced HUVECs angiogenesis. (**C**) mRNA levels of angiogenesis factors (ANG1, FILK1, and VEGF) and angiogenesis inhibitors (VASH1 and TSP1) were analyzed by qRT-PCR. The data are shown as the mean  ±  SD (* *p*  <  0.05, ** *p*  <  0.01).

**Table 1 cells-11-02568-t001:** The list of PCR primers.

Mouse GAPDH	F:5′-GTGTTCCTACCCCCAATGTG-3′	R:5′-CATCGAAGGTGGAAGAGTGG-3′
Mouse VEGF	F:5’-GTGGACATCTTCCAGGAGTA-3’	R:5’-TCTGCATTCACATCTGCTGT-3’
Mouse VCAM	F:5’-GTTCCAGCGAGGGTCTACC-3’	R:5’-AACTCTTGGCAAACATTAGGTGT-3’
Mouse IL-1	F:5′-TCCTGTGTAATGAAAGACGGC-3′	R:5′-TGCTTGTGAGGTGCTGATGTA-3′
Mouse IL-6	F:5′-GGGACTGATGCTGGTGACAA-3′	R:5′-TCCACGATTTCCCAGAGAACA-3′
Mouse TNF-α	F:5′-GAACTGGCAGAAGAGGCACT-3′	R:5′-CATAGAACTGATGAGAGGGAGG-3′
Mouse MCP-1	F:5′-ACTGCACCCAAACCGAAGTC-3′	R:5′-TGGGGACACCTTTTAGCATCTT-3′
Human GAPDH	F:5′-ATGGGGAAGGTGAAGGTCG-3′	R:5′-GGGGTCATTGATGGCAACAATA-3′
Human ANG1	F:5′-CAGACTGCAGAGCAGACCAGAA-3′	R:5′-CTCTAGCTTGTAGGTGGATAATGAATTC-3′
Human FLK1:	F:5’-GACTTCCTGACCTTGGAGCATCT-3’	R:5’-GATTTTAACCACGTTCTTCTCCGA-3′
Human VASH1	F:5′-AACTACTTCCGCCACATCGT-3′	R:5′-GGCGGCTTGTACATCAGGTC-3′
Human TSP1	F:5′- UCCUUCUAGGUGGCCUCAGAC-3′	R:5′-CATTGGAGCAGGGCATGATGG-3′
Human VEGF	F:5′-GCAGAATCATCACGAAGTGGTG-3′	R:5′-TCTCGATTGGATGCAGTAGCT-3′
Human VCAM	F:5′-GTAAAAGAATTGCAAGTCTACATATCAC-3′	R:5′-GATGGATTCACAGAAATAACTGTATTC-3′
Human ICAM	F:5′-AACCAGAGCCAGGAGACACTG -3′	R:5′-GCGCCGGAAAGCTGTAGATG-3′
Human IL-1	F:5′-ATGATGGCTTATTACAGTGGCAA -3′	R:5′-GTCGGAGATTCGTAGCTGGA-3′
Human IL-6	F:5′-ACTCACCTCTTCAGAACGAATTG-3′	R:5′-CCATCTTTGGAAGGTTCAGGTTG-3′
Human TNF-α	F:5′-CTATCTGGGAGGGGTCTTCC-3′	R:5′-GGTTGAGGGTGTCTGAAGGA-3′
Human MCP-1	F:5′-CAGCCAGATGCAATCAATGCC-3′	R:5′-TGGAATCCTGAACCCACTTCT-3′

## Data Availability

The data and materials used to support the findings of this study are available from the corresponding author upon request.

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
