# Peer review of "Exosomes Derived from Adipose Mesenchymal Stem Cells Promote Diabetic Chronic Wound Healing through SIRT3/SOD2"

_cells, 2022, doi:10.3390/cells11162568_

Round 1

Reviewer 1 Report

The manuscript by Zhang Y et al. reports on the role of exosomes from stem cells derived from adipose tissue on the oxidative stress and inflammation mechanisms caused by wounds in diabetic mice.

They found that the treatment with exosomes restored the expression of the gene Sirtuin 3 (SIRT3) that was reduced by high glucose environment. As a result, exosome treatment improved oxidative stress and inflammation in wounds, promoted angiogenesis, and accelerated diabetic wound healing.

The topic is of interest, the methodology is appropriate, the results are promising.

Comments

-Title and text: please avoid the reiteration of the word "derived" in the same sentence.

-Specify acronyms in figures.

-Page 11. lines 328-341. These sentences are supposed to integrate the introduction paragraph.

Reviewer 2 Report

The authors investigate the effect of adipose-derived mesenchymal stem cell-derived exosomes (ADSC-exos) on diabetic wounds and the mechanism underlying this effect. The results showed  that ADSC-exos could improve oxidative stress and secretion of inflammatory cytokines in diabetic  wounds, thereby increasing periwound vascularization and accelerating wound healing.

Very interesting work! I have the following questions:

a) How was the response of cellular infiltrate?

b) How about mast cells?

c) How about macrophages?

d) How about fibroblasts?

Reviewer 3 Report

The article describes the effect of exosomes isolated from adipose-derived mesenchymal stem cells on diabetic wound healing. The study was very interesting and well conducted, both in vitro and in vivo. However, the illustrations and graphs are poorly described, they are not clearly presented. Some methods are not sufficiently detailed.

Minor issues:

1. In Materials and Methods section, 2.1 , the phenotype of ADSC was evaluated with anti CD90 and CD105 antibodies (line 82), did the Authors check anti CD73, the third canonical marker of MSC? Similarly, did you check chondrogenesis, apart from osteo- and adipogenesis (line 83)?

2. There are some abbreviations not explained in Materials and Methods section: CCK-8 (line 142), MMP (line 158), JC-1 (line 159), AC-SOD2 (line 174). Please clarify.

3. In 2.11 section (Western blot analysis) the Authors did not write how much protein was loaded into each well. This is important information, because the Authors stated in the next paragraph (Results, 3.1, line 195) that they observe high expression of CD63 and CD9 in Western blot analysis.

4. Fig. 1B – the histograms lack isotypic control, it is impossible to say whether the negative or positive expression is seen. Fig. 1C – the scale bar is completely invisible.

5. In section 3.2 FBG level in db/db mice exceeded 16.7 mmol/L but it is no information what is the normal FBG level (lines 207-208). Line 217 … blood vessel density was significantly higher than 217 that in the control group… This is an undocumented statement, because there is no statistical analysis.

6. Fig. 2B - what the Authors understand by “wound healing ratio (%)” on Y axis? This is not explained either in Figure legend or in Materials and Methods section.

7. Section 3.3 – again, there are some abbreviations not explained: ANG1, FLK1, VASH1, TSP1 (lines 248-249). Maybe the Authors should include a new section with all abbreviations used in the text?

8. Section 3.4 – how the Authors document the protective effect of exosomes on mitochondria (lines 270-271) assessed by JC-1 staining? In Fig. 4B there are staining of JC1 monomers and aggregates but there is no explanation what it means. The Western blot results described in the text (lines 276-281) are not visible on the blot (Fig. 4D). I can’t see the difference in SOD2 and SIRT3 expression band, maybe it is the problem with image resolution.

Round 2

Reviewer 2 Report

The authors have answered correctly to my questions